# EGCG, a Green Tea Compound, Increases NO Production and Has Antioxidant Action in a Static and Shear Stress In Vitro Model of Preeclampsia

**DOI:** 10.3390/antiox13020158

**Published:** 2024-01-26

**Authors:** Mariana Bertozzi-Matheus, Thaina Omia Bueno-Pereira, Priscila Rezeck Nunes, Valeria Cristina Sandrim

**Affiliations:** Department of Biophysics and Pharmacology, Institute of Biosciences, Sao Paulo State University (UNESP), Botucatu 18618-689, SP, Brazil; mariana.bertozzi@unesp.br (M.B.-M.); thaina.omia@unesp.br (T.O.B.-P.); priscila.nunes@unesp.br (P.R.N.)

**Keywords:** preeclampsia, endothelial dysfunction, NO, antioxidant, EGCG, green tea

## Abstract

Preeclampsia (PE) is a gestational hypertensive disease characterized by endothelial dysfunction. Epigallocatechin-3-gallate (EGCG), the main compound in green tea, is a promising therapeutic target for the disease. By activating eNOS, EGCG increased NO production and exerted an important antioxidant action, but its specific impact in the context of PE remains understudied. The aim of this study is to evaluate the effects of EGCG on endothelial function in static and shear stress in in vitro models of PE. Endothelial cells were incubated with healthy (HP) and preeclamptic (PE) pregnant women’s plasma, and the latter group was treated with EGCG. Additionally, NOS (L-NAME) and PI3K protein (LY249002) inhibitors were also used. The levels of NO, ROS, and O2^•−^ were evaluated, as well as the antioxidant potential. These investigations were also carried out in a shear stress model. We found that EGCG increases the NO levels, which were reduced in the PE group. This effect was attenuated with the use of L-NAME and LY249002. Furthermore, EGCG increased the antioxidant capacity of PE, but its action decreased with LY294002. In cells subjected to shear stress, EGCG increased nitrite levels in the PE group and maintained its action on the antioxidant capacity. This is the first study of the effects of EGCG in this experimental model, as well as the investigation of its effects along with shear stress. Our findings suggest that EGCG improves parameters of endothelial dysfunction in vitro, making it a promising target in the search for treatments for the disease.

## 1. Introduction

Preeclampsia (PE) is a pregnancy disorder characterized by hypertension accompanied by proteinuria and/or target organ dysfunction [1,2]. PE affects an estimated 2–8% of pregnant women globally, and this percentage is higher in underdeveloped countries [1]. Poor remodeling of uteroplacental spiral arterioles, inadequate placentation, and insufficient placental perfusion are known to lead to hypoxia and oxidative stress [3,4,5]. This condition results in the release of oxidant, antiangiogenic, and inflammatory factors into the maternal circulation, causing significant endothelial dysfunction, a crucial factor in the development of maternal syndrome [2,6,7]. An important feature of pregnant women with PE is the decrease in Nitric Oxide (NO) bioavailability [8], an important vasodilator produced by endothelial nitric oxide synthase (eNOS), which can also become dysfunctional in these women due to factors such as uncoupling or inactivation [6,8,9].

At present, research groups look for compounds or drugs that can help in the treatment or prevention of PE, a disease that has no cure [2,10,11]. The antihypertensive and anti-inflammatory properties of foods and beverages derived from natural sources and rich in polyphenols have been established as significant in the context of cardiovascular and metabolic diseases [12,13,14]. Among these beverages are teas, which are accessible and generally do not have adverse effects or restrictions [15,16]. In this context, we can cite green tea and its main compound, epigallocatechin-3-gallate (EGCG), which has an important vasodilator and antioxidant properties that can improve endothelial function. In addition to directly sequestering reactive oxygen species (ROS), EGCG increases the bioavailability of NO through the activation of eNOS via PI3K and may help in the treatment of several cardiovascular and metabolic diseases [17,18,19,20]. However, studies showing the effect of EGCG in the PE context are limited, and none use the experimental model of this study [21,22,23].

Considering the relevance of the endothelium and the release of negative factors into the maternal circulation as the key points for the development of PE, this study utilized an in vitro model of endothelial cells (EC) incubated with the plasma of healthy and preeclamptic women [24,25,26,27,28]. We evaluated the effect of these samples on the cells in the presence or absence of EGCG. To investigate the action mechanism of the treatment, PI3K and NOS inhibitors were also used. In addition, considering the PE pathophysiology complexity and the limitation of in vitro studies in reproducing what occurs in the organism of these pregnant women, we employed a methodology that mimics the shear stress force present in the circulatory system to create a model that better simulates the effects of blood flow on the endothelium [29]. As shear stress acts on endothelial function and there are few studies investigating this action in the setting of PE, this in vitro methodology becomes another differential of our study.

Thus, the present study aims to assess the impact of EGCG on endothelial function in vitro, including the production of NO, oxidizing agents, and antioxidant response. We investigate the potential benefits of EGCG in this static and shear stress cell culture model and consider its implications for PE.

## 2. Materials and Methods

### 2.1. Participants Selection

Blood samples were collected from healthy pregnant women (*n* = 10) and preeclamptic women (*n* = 10) at the Ribeirão Preto Clinical Hospital (Ribeirão Preto, Brazil) in tubes containing heparin. Samples were centrifuged at room temperature for 10 min at 1200× *g* to obtain plasma. Aliquots of 1000 µL were separated and stored at −80 °C until use in incubation with human umbilical vein endothelial cells (HUVECS). The study was approved by the Research Ethics Committee of the Faculty of Medicine of Ribeirão Preto, Brazil (CAAE 37738620.0.0000.5440, 19 October 2020), following the principles of the Declaration of Helsinki. All participants signed the free and informed consent form. Preeclampsia diagnostic criteria were defined by the American College of Obstetricians and Gynecologists (ACOG), and women pregnant with twins, as well as women with chronic arterial hypertension, autoimmune, renal, or hepatic diseases, were excluded from the study. Pregnant women in the HP and PE groups with similar ages and gestational weeks at the time of collection were chosen.

### 2.2. Cell Culture

Human umbilical vein endothelial cells (HUVECS), lineage EA.hy 926, obtained from the Rio de Janeiro Cell Bank (BCRJ, Duque de Caxias, RJ, Brazil), were used in this study. The cells were cultivated in culture flasks at 37 °C with 5% CO_2_ using a medium supplemented with 10% (*v*/*v*) fetal bovine serum (FBS), 50 μg/mL of penicillin, 100 μg/mL of streptomycin and 0.5 μg/mL of amphotericin B until they reached 80–90% confluency. For the experiments, the EC were seeded in culture plates in the same medium until they reached the necessary confluency. Subsequently, the culture medium was removed, and the cells were washed with PBS and incubated in culture medium without FBS and with 10% (*v*/*v*) of HP and PE plasma for 24 h at 37 °C in 5% CO_2_. In the treated cells, the same procedures were performed, but in the presence or absence of EGCG (Cayman Chemical^®^, Ann Arbor, MI, USA) (100 μM) during the last 60 min of incubation and L-NAME (Cayman Chemical^®^, Ann Arbor, MI, USA) (100 μM), NOS-inhibiting substance, and LY294002 (Cayman Chemical^®^, Ann Arbor, MI, USA) (30 μM), a PI3K inhibitor, 30 min before incubation with plasma. These procedures were performed prior to all experiments described below.

### 2.3. Intracellular NO

Intracellular NO analysis was performed using a reagent that diffuses into cells and reacts with intracellular NO before being deacetylated by esterases and becoming highly fluorescent [30]. To perform this assay, after plating and incubating the cells, the culture supernatant was removed from the wells for the subsequent addition of PBS (95 µL) and incubation for 30 min. Thus, the DAF-FM solution (2.5 µM) (Invitrogen, Thermo Fisher Scientific, Waltham, MA, USA) was added to each well, and the measurement was conducted after one hour using the Synergy 4 plate reader (BioTek, Winooski, VT, USA) with 485 nm excitation and 520 nm emission.

### 2.4. ROS Quantification

2′,7′-dichlorodihydrofluorescein (DCFH) is a reduced fluorescein reagent used as an indicator of reactive oxygen species in cells. After cleavage of its acetate groups by intracellular esterases and oxidation, this non-fluorescent substance is converted into 2′,7′-dichlorofluorescein (DCF), which is read by fluorescence [31]. For this, after plating and incubating the cells, the supernatant from all wells was discarded, and 200 µL of DCFH-DA solution (25 µM) (Cayman Chemical^®^, Ann Arbor, MI, USA) was added to each well and incubated for 30 min at 37 °C. After this period, the supernatant was discarded, and the wells were washed once with PBS. For the reading, 200 µL of PBS was added to each well and was read in the Synergy 4 plate reader (BioTek, Winooski, VT, USA) with 485 nm excitation and 520 nm emission.

### 2.5. Superoxide Quantification

Superoxide quantification is performed using the DHE (Dihydroethidium) probe. When oxidized by the O2^•−^ anion, the probe is converted into 2-hydroxyethidium (2-OH-E+), a fluorescent compound read at 500–530 nm of excitation and 590–620 nm of emission [32]. To perform this assay, after plating and incubating the cells, the supernatant containing the plasma and treatments was discarded for the later addition of PBS in the wells. Then, 20 µM of DHE reagent (Sigma-Aldrich, St. Louis, MO, USA) was added to the wells for 30 min at 37 °C in 5% CO_2_. The evaluation of superoxide production was quantified by measuring the fluorescence using the Synergy 4 microplate reader (BioTek, Winooski, VT, USA) at 518 nm of excitation and 620 nm of emission.

### 2.6. Antioxidant Capacity

The antioxidant response of the samples was evaluated using the Ferric Reducing Antioxidant Power (FRAP) (ability to reduce iron Fe^3+^ to Fe^2+^) in the cell culture supernatant [33]. For this assay, 80 μL of the samples were pipetted into test tubes containing the working solution consisting of acetate buffer (pH 3.6), 10 mM of 2,4,6-tri(2-pyridyl)-1,3,5-triazine (TPTZ), and 20 mM of ferric chloride (FeCl_3_·6H_2_O). The tubes were then incubated at 37 °C for 15 min. After the incubation period, the samples, along with a standard curve of ferrous sulfate (Fe_2_SO_4_·7H_2_O), were transferred to transparent plates for measurement using the Synergy 4 microplate reader (BioTek, Winooski, VT, USA) at an absorbance of 593 nm.

### 2.7. Shear Stress

To assess the impact of shear stress on endothelial cells, EC were cultured and seeded on modified 100 mm culture plates. These plates had a 60 mm plate attached to the center and had been previously sterilized. The culture plates were maintained at 37 °C with 5% CO_2_ until the cells adhered and reached the desired confluence. Subsequently, laminar fluid flow was applied to the culture plates using the AO-330D shaking equipment (Gehaka, São Paulo, SP, Brazil) at a predetermined rotation frequency. This generated a constant shear stress ranging from 6–40 dynes/cm^2^ (equivalent to rotations per second: 199 rpm), which represented physiological conditions. After 48 h in the shaker, the cells were incubated with the PE plasma and rotated again for 24 h, totaling 72 h of rotation. EGCG (Cayman Chemical^®^, Ann Arbor, MI, USA) (100 μM) was added in the last 60 min. After this period, the NO production was assessed by detecting nitrite levels in the culture supernatant. Additionally, an iron reduction (as described above) was conducted to analyze the antioxidant capacity of the supernatant.

### 2.8. Nitrite Supernatant Quantification

The measurement of nitrite production serves as an indirect indicator of NO production. This assessment involves the conversion of sulfanilic acid to a diazonium salt through its reaction with nitrite in an acidic solution. The formed salt couples with N-(1-naphthyl) ethylenediamine, resulting in the generation of a colored compound proportional to the nitrite content in the sample. For this evaluation, the Griess Reagent kit (Invitrogen, Thermo Fisher Scientific, Waltham, CA, USA) was employed following the manufacturer’s instructions. In a clear 96-well plate, 75 µL of the sample was combined with the necessary reagents (10 µL) for the aforementioned reaction. Additionally, a nitrite standard curve was prepared and measured using a spectrophotometer at an absorbance of 548 nm.

### 2.9. Statistical Analyses

Replicates of five per group, combined with treatments (plasma, EGCG, and inhibitors), were performed in each experiment. To compare the two groups, we performed multiple *t*-tests or Mann–Whitney test for non-parametric values. When comparing three or more groups, we used a one-way ANOVA test followed by the Dunnet’s multiple comparisons test or the Kruskall–Wallis test followed by the Dunn’s multiple comparisons test for non-parametric values. Results are expressed in means ± SEM. Statistical analyses were performed using GraphPad Prism 8.0 (GraphPad Software, San Diego, CA, USA), and for all tests, a *p*-value ≤ 0.05 (two-tailed) was considered significant.

## 3. Results

### 3.1. EGCG Increases NO Levels in PE Group via PI3K and eNOS

EC incubated with PE plasma produced less NO than cells incubated with HP plasma (PE: 82.26 ± 4.89 vs. HP: 100.00 ± 3.67), and EGCG treatment recovered these levels in the first group (PE: 82.26 ± 4.89 vs. PE+EGCG: 103.83 ± 8.31, respectively) (Figure 1A). In addition, the inhibitor of NOS, L-NAME, and PI3K protein, LY294002, reduced the EGCG action (59.32 ± 6.54 and 67.23 ± 1.82 vs. 103.83 ± 8.31, respectively) (Figure 1B).

### 3.2. EGCG Does Not Alter ROS and O2^•−^ Levels

Our data show that the PE group exhibited lower levels of ROS (PE: 89.83 ± 2.98 vs. HP: 100.00 ± 2.98) (Figure 2A) and similar O2^•−^ levels in comparison to the HP group (PE: 100.87 ± 3.41 vs. HP: 99.36 ± 1.83) (Figure 2B), while EGCG treatment did not affect these parameters. L-NAME and LY294002 did not change the action of EGCG.

### 3.3. EGCG Increases Antioxidant Potential

The antioxidant potential between the HP and PE group is similar (HP: 100.00 ± 2.14 vs. PE: 116.31 ± 15.15). However, EGCG considerably increases these levels in the PE group (PE: 116.31 ± 15.15 vs. PE + EGCG: 289.23 ± 4.73) (Figure 3A). L-NAME did not change the action of EGCG, but LY294002 attenuates the phytochemical action in the antioxidant capacity (PE + EGCG: 289.23 ± 4.73 vs. PE + EGCG + LY294002: 232.83 ± 4.84) (Figure 3B).

### 3.4. EGCG Increases Nitrite Levels and Has Antioxidant Capacity in Shear Stress

To investigate whether shear stress interferes with our treatment, we evaluated the nitrite levels in the supernatant of cells treated with plasma from PE patients with or without EGCG and subjected to shear stress. Our results showed that EGCG significantly increased nitrite levels compared to the untreated PE group (PE: 100.00 ± 5.00 vs. PE + EGCG: 273.09 ± 21.82) (Figure 4A).

Regarding the antioxidant action of EGCG, we observed that it remained effective even in cells exposed to shear stress. As shown in Figure 4B, the antioxidant capacity of the group treated with EGCG is higher compared to the untreated group (PE: 158.28 ± 5.49 vs. PE + EGCG: 473.22 ± 8.18).

## 4. Discussion

In this work, we observed that EGCG increased NO levels and antioxidant capacity in EC treated with PE plasma. This is one of the few works that investigate the action of EGCG in the PE field and the only one in this in vitro model that also considers the action of shear stress on cells.

Lower levels of NO in the organism of pregnant women with PE have already been described in several studies [8,9,34]. The decrease in this vasodilator contributes to the characteristic hypertension of the disease [9,35,36]. Our results are in line with these works, showing that endothelial cells incubated with PE plasma have lower levels of intracellular NO compared to cells incubated with HP plasma. Moreover, treatment with EGCG restores these levels in the PE group.

It is known that this phytochemical has an important action on eNOS, the enzyme responsible for NO production [22,37]. By activating the PI3K/Akt/eNOS pathway, leading to the enzyme activation, EGCG is related to an increase in NO production and bioavailability [17,19,20]. To investigate the mechanism of EGCG on this pathway, we used the NOS inhibitor (L-NAME) and the PI3K protein inhibitor (LY294002) with our treatment. The results demonstrated that these inhibitors considerably reduced the action of the phytochemical, demonstrating the importance of this pathway in the EGCG action on the NO bioavailability.

A limited number of studies have explored the mechanism preceding PI3K activation by EGCG. However, one particular study has proposed the involvement of Src family kinase Fyn [19]. The study showed that siRNA knockdown of Fyn inhibited eNOS phosphorylation by EGCG. Furthermore, this investigation suggested that the PI3K activation could occur both by direct binding of Fyn in its p85 regulatory domain and by phosphorylation of intracellular substrates, mainly Gab-1, which binds and activates PI3K [19].

In addition to the NO decrease, some studies point to an increase in free radicals such as ROS and O2^•−^ in PE [3,38,39]. However, these results are controversial, and similar levels of oxidative stress markers have already been found in the plasma of pregnant women with PE [40,41]. Likewise, when evaluating ROS and O2^•−^ levels, our results demonstrate that O2^•−^ levels are similar between the HP and PE groups, and the total ROS levels are even lower in the PE group when compared to the HP group, while EGCG does not change these levels. In this way, we hypothesize the presence of a compensatory mechanism responding to oxidative stress since previous findings by our research team showed similar or even increased levels of antioxidant potential in the PE group when compared to the HP group [27,42]. This work is in agreement with these results, and similar levels of this parameter were found between the two groups. In addition, EGCG considerably increases this response, given its significant antioxidant properties.

The L-NAME did not change the EGCG action on the antioxidant capacity. However, the use of LY294002 decreased the phytochemical effects. When researching alternative pathways to PI3K/AKT/eNOS that could be involved in the EGCG antioxidant effects, we found some studies that report the action of the nuclear factor Nrf2 [43,44,45]. Some groups point out that EGCG can activate this factor directly or via PI3K, which would explain the decrease in the EGCG antioxidant action after the use of LY294002 [43,45]. After activation, Nrf2, in the nucleus, binds to the antioxidant response element (ARE), triggering the transcription of several antioxidant genes [44]. Moreover, EGCG exhibits a well-documented antioxidant impact, having the capability to directly neutralize oxidizing agents [46]. The LY294002 moderate effect may be due to this direct antioxidant action of EGCG. Nevertheless, further investigations regarding the role of EGCG in this pathway should be performed. An overview of all the mechanisms discussed is presented in Figure 5.

PE is a condition exclusive to human pregnancy, making the use of animals as experimental models limited. Additionally, due to its multifactorial nature and complex pathophysiology, in vitro models do not accurately replicate the true disease environment. Thus, methodologies are required to narrow the disparity between these models and the physiological conditions of pregnant women affected by PE. For example, in blood vessels, shear stress modulates arterial function in endothelial cells, an effect not achievable in static cell culture. The action of laminar shear stress, unidirectional and constant, has a protective effect on cells, regulating eNOS and coagulation, promoting vasodilation, and favoring antioxidant action [47].

Despite the limited number, studies indicate that the action of shear stress interferes in the PE context. One study showed a lower release of NO, mediated by shear stress in myometrial arteries of pregnant women with PE when compared to normotensive pregnant women [48], while in vitro studies pointed out that the application of shear stress had an effect on the production of nitrite, increasing the levels of this substance both in the supernatant of cells incubated with PE plasma [49] and in endothelial cells from the decidua of pregnant women with PE [50].

With the goal of replicating the effects of shear stress experienced by our blood vessels, we applied a rotational force to simulate unidirectional blood flow and the resulting mechanical stress on endothelial cells [29,51]. Although the shear stress acquired with the applied rotational force is equivalent to that of a normotensive person, we observed that EGCG exhibited consistent effects even under these conditions, increasing nitrite levels in the supernatant of cells incubated with PE plasma (at levels even higher than the increase in NO in the static culture) and the antioxidant capacity of the PE group treated with EGCG.

Nowadays, several research groups have focused on plant-derived and natural compounds for treating and preventing PE [26,27,42]. EGCG, the primary compound found in green tea, has demonstrated beneficial properties in the context of cardiovascular and metabolic diseases [18,52,53]. Despite limited research on the action of EGCG in PE, the existing studies show promising results [22]. In one study, the administration of EGCG capsules to PE patients using hydralazine, an antihypertensive drug for controlling hypertensive crises, reduced the necessary doses of the drug, as well as the interval between doses [23]. Additionally, an in vitro study of EC exposed to hypoxia demonstrated that EGCG improved endothelial dysfunction parameters and the anti-angiogenic state [21].

Some limitations of the study include the concentration of EGCG used in vitro, which is higher than plasma levels achieved in pregnant women if they consumed EGCG [17]. Regarding this, a study showed that 1 µM of the phytochemical was responsible for causing vasodilation in the aortic rings of rats [17], in addition to the clinical study that showed that EGCG capsules consumed by PE pregnant women had an effect in clinical practice [23]. Furthermore, the consumption of teas during pregnancy should be considered with caution. This, however, does not exclude the isolated EGCG beneficial and safe effects.

In summary, our results indicate that both in static culture and in cells subjected to shear stress, EGCG improves parameters of endothelial function and attenuates oxidative stress in cells exposed to the plasma of PE pregnant women.

## 5. Conclusions

Our analysis shows that EGCG increases NO levels through the activation of the PI3K/AKT/eNOS pathway and enhances antioxidant capacity in cells incubated with PE plasma. These beneficial effects are also observed in cells exposed to shear stress. Thus, we suggest that EGCG acts on endothelial function, offering potential therapeutic insights for PE treatment.

## Figures and Tables

**Figure 1 antioxidants-13-00158-f001:**
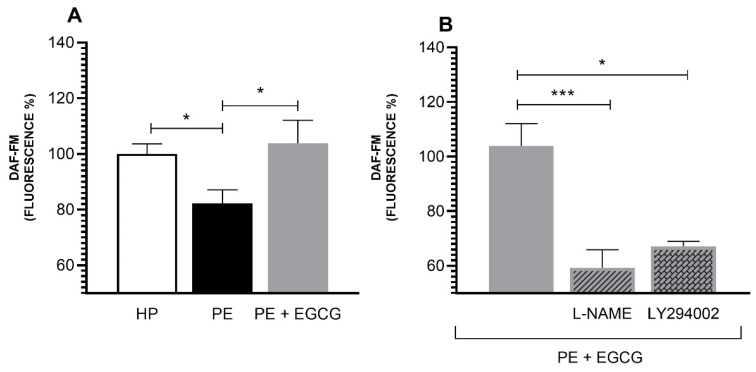
Intracellular Nitric Oxide Levels. NO fluorescence intensity was measured by DAF-FM. (**A**) The PE group had lower NO levels when compared to the HP group, and the EGCG treatment (100 μM) recovered these levels. (**B**) By adding L-NAME (100 μM) and LY294002 (30 μM), NOS, and PI3K inhibitors, respectively, the action of EGCG was reduced. NO levels are presented in fluorescence percentage based on the HP group. Data are presented as mean ± SEM, and comparisons between groups were assessed by a Kruskal–Wallis test followed by a Dunn’s Multiple Comparison Test. * *p* < 0.05; *** *p* < 0.001.

**Figure 2 antioxidants-13-00158-f002:**
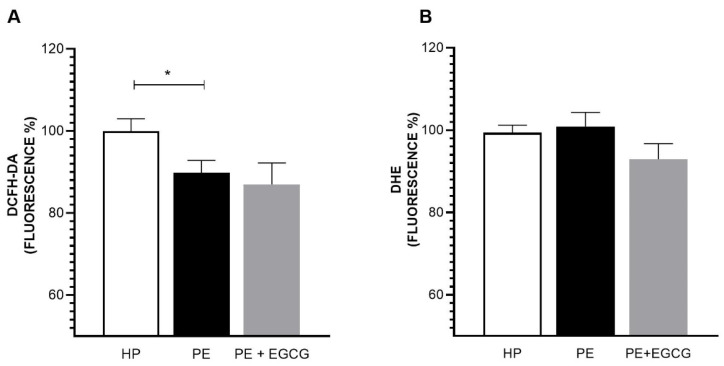
Levels of total ROS and O2^•−^. Total ROS was measured by DCFH-DA and O2^•−^ by DHE. (**A**) The reactive oxygen species levels in the PE group were lower when compared to the HP group, and the EGCG (100 μM) treatment did not change these levels. (**B**) Superoxide levels were similar in all groups. ROS and O2^•−^ levels are presented in fluorescence percentages based on the HP group. Data are presented as mean ± SEM, and comparisons between groups were assessed by (**A**) One-Way ANOVA test followed by Dunnet’s Multiple Comparison Test and (**B**) Kruskal–Wallis test followed by Dunn’s Multiple Comparison Test. * *p* < 0.05.

**Figure 3 antioxidants-13-00158-f003:**
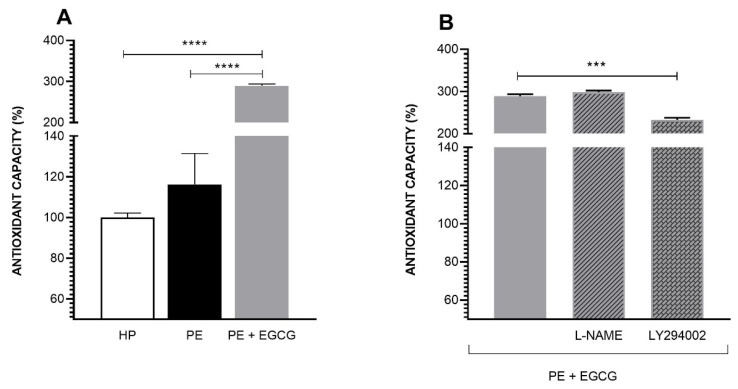
Antioxidant capacity. The antioxidant capacity was measured by FRAP assay. (**A**) The antioxidant potential is similar between HP and PE groups, and the EGCG (100 μM) treatment increases this potential in PE. (**B**) LY294002 (30 μM) but not L-NAME (100 μM) attenuated the EGCG action. The results are presented in absorbance percentage based on the HP group. Data are presented as mean ± SEM, and comparisons between groups were assessed by a one-way ANOVA test followed by Dunnet’s Multiple Comparison Test. *** *p* < 0.001; **** *p* < 0.0001.

**Figure 4 antioxidants-13-00158-f004:**
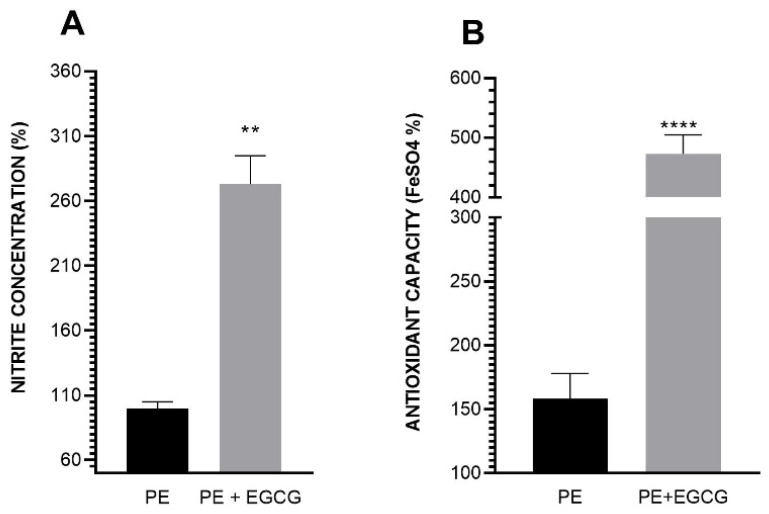
Nitrite levels and antioxidant capacity in cell culture supernatant exposed to shear stress. EC were exposed to shear stress for 72 h, with PE plasma in the last 24 h and in the presence or absence of EGCG in the last 60 min. Then, the supernatant was collected for Griess and FRAP assays. (**A**) The nitrite level was higher in the group treated with EGCG (100 μM). (**B**) Antioxidant capacity was also higher in the same group. The results are presented in absorbance percentage based on the HP group, and data are presented as mean ± SEM. Comparisons between groups were assessed using an unpaired *t*-test. ** *p* < 0.01; **** *p* < 0.0001.

**Figure 5 antioxidants-13-00158-f005:**
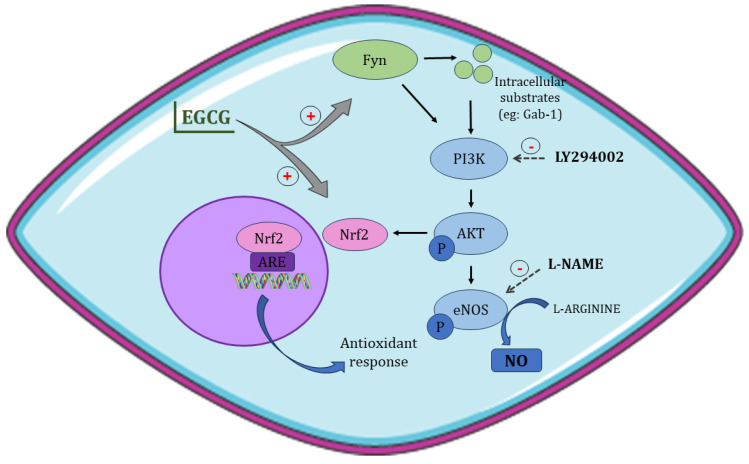
Proposed mechanisms of EGCG on endothelial cells. By triggering the Fyn protein activation, EGCG initiates the PI3K/Akt/eNOS pathway. This cascade involves a series of phosphorylation events, starting with Akt phosphorylation and culminating in eNOS activation. Consequently, this sequence leads to an increase in NO production. Furthermore, EGCG also triggers the activation of the Nrf2 factor, influencing the antioxidant response. This is achieved through the interaction of Nrf2 with ARE, which subsequently stimulates the synthesis of numerous antioxidant enzymes.

## Data Availability

The data presented in this study are available in the article.

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
