# Peer review of "EGCG, a Green Tea Compound, Increases NO Production and Has Antioxidant Action in a Static and Shear Stress In Vitro Model of Preeclampsia"

_antioxidants, 2024, doi:10.3390/antiox13020158_

Round 1

Reviewer 1 Report

Comments and Suggestions for Authors

1)      Line 97, Please clarify, Do the inhibitors (L-NAME and LY294002 ) were added 30 minutes before incubation with  plasma or 30 min before the incubation with EGCG? .

2)      In results to add to the mean value also the SEM.

3)      Please, double check the significativity in Fig 1B : PE + EGCG vs  PE + EGCG + LY 294002

4)      Line 214 antioxidant activity, please to be uniform to use antioxidant capacity

5)      In your system, Did shear stress increased  NO production and/or antioxidant capacity? If you have the data please to add them in the fig. 4.

6)      Line 282-288, the authors argue about the limit of animal model for the PE, but the author should refer to the limit of colture cell. However, It looks out of the text contest.

Author Response

We appreciate the reviewer for their valuable contributions. Please see the attachment with the responses

Reviewer 2 Report

Comments and Suggestions for Authors

This paper explores the effects of EGCG on endothelial function in HUVECs incubated with plasma from pre-eclamptic or control pregnant women, to determine whether it has any beneficial effects on nitric oxide production and anti-oxidant capacity. Some suggestions to improve the paper are below:

Figure 1. It would be useful to see whether EGCG increases the NO levels in HP conditions. Also, fluorescence was measured every 10 minutes, however only one timepoint is shown. The timepoint should be stated in the legend, however it would be better to show the kinetic data.

Figure 3. LY294002 only moderately reduced the antioxidant capacity mediated by EGCG, it is not reversed back to PE levels alone. This is not discussed or mentioned. What other Pi3K independent mechanisms may be responsible

Comment should be made in the discussion on how the shear-stress conditions in this model compare with shear stress levels in HP and PE? Is the model representing shear stress in patients with hypertension (increased shear stress) or normal preganancies?

Figure 5 – The title of this figure should be changed to reflect that this is a proposed mechanism since the study has not demonstrated a role for AKT or Nrf. For example ‘Proposed mechanism of action of EGCG on endothelial cells’

Minor changes:

L18 – change ‘founded’ to ‘found’

L53 –‘Considering the endothelium…..’ this sentence requires restructuring to improve the flow

L57 – ‘To investigate the action mechanism of the treatment…..’  change to ‘To investigate the mechanism of action of the treatment…’

L99 Intracellular spelt incorrectly

L104 should this be 2.5µM?

L237 ‘organisms’ is the incorrect word

L251 – This paragraph should be simplified for example : A limited number of studies have explored the mechanism preceding PI3K activation by EGCG. However, one study has proposed the involvement of Src family kinase Fyn, [19]. The study showed that siRNA knockdown of Fyn inhibited eNOS phosphorylation by EGCG. Furthermore, the investigation suggested that the PI3K activation could occur both by direct binding of Fyn in its p85 regulatory domain, and by phosphorylation of intracellular substrates, mainly Gab-1, which binds and activates PI3K [19]

Src family proteins should be Src family kinases and Fyn should just be referred to as Fyn not ‘the Fyn protein’.

L312 change ‘than that available in the pregnant women organism if they consumed EGCG’ to ‘than plasma levels achieved in pregnant women if they consumed EGCG’

Comments on the Quality of English Language

Some minor editing required.

Author Response

(The authors gave the same response as above.)
